# Regular Exercise Rescues Heart Function Defects and Shortens the Lifespan of *Drosophila* Caused by dMnM Downregulation

**DOI:** 10.3390/ijerph192416554

**Published:** 2022-12-09

**Authors:** Wenzhi Gu, Qiufang Li, Meng Ding, Yurou Cao, Tongquan Wang, Shihu Zhang, Jiadong Feng, Hongyu Li, Lan Zheng

**Affiliations:** Key Laboratory of Physical Fitness and Exercise Rehabilitation of Hunan Province, Hunan Normal University, Changsha 410012, China

**Keywords:** regular exercise, dMnM, cardiac function, IFM, lifespan

## Abstract

Although studies have shown that myomesin 2 (MYOM2) mutations can lead to hypertrophic cardiomyopathy (HCM), a common cardiovascular disease that has a serious impact on human life, the effect of MYOM2 on cardiac function and lifespan in humans is unknown. In this study, dMnM (MYOM2 homologs) knockdown in cardiomyocytes resulted in diastolic cardiac defects (diastolic dysfunction and arrhythmias) and increased cardiac oxidative stress. Furthermore, the knockdown of dMnM in indirect flight muscle (IFM) reduced climbing ability and shortened lifespan. However, regular exercise significantly ameliorated diastolic cardiac dysfunction, arrhythmias, and oxidative stress triggered by dMnM knockdown in cardiac myocytes and also reversed the reduction in climbing ability and shortening of lifespan caused by dMnM knockdown in *Drosophila* IFM. In conclusion, these results suggest that *Drosophila* cardiomyocyte dMnM knockdown leads to cardiac functional defects, while dMnM knockdown in IFM affects climbing ability and lifespan. Furthermore, regular exercise effectively upregulates cardiomyocyte dMnM expression levels and ameliorates cardiac functional defects caused by *Drosophila* cardiomyocyte dMnM knockdown by increasing cardiac antioxidant capacity. Importantly, regular exercise ameliorates the shortened lifespan caused by dMnM knockdown in IFM.

## 1. Introduction

Cardiovascular disease (CVD) is the leading cause of death worldwide, with more than four million deaths recorded in Europe yearly (46% of all deaths) [1,2]. Hypertrophic cardiomyopathy (HCM) is the most common inherited cardiovascular disease. HCM affects 1 in 500 people and is the leading cause of heart failure and sudden death [3,4,5]. Although recent studies have identified mutations in the myojunction gene MYOM2 in HCM patients [6], these patients have no mutations in the 12 most common pathogenic genes, such as MYH7, MYBPC3, etc. Therefore, MYOM2 may serve as a new candidate gene for HCM pathogenesis.

Myomesin 2 (MYOM2), also known as M-protein, is located in the M-band of the transverse muscle [7] and belongs to the myomesin gene family. To date, only one myomesin-2 isoform has been identified, with major differences at the protein terminus. This isoform has a much shorter N-terminal than myomesin 1, and its C-terminal contains a unique, strongly basic multi-proline sequence with no similarity to myomesin 1 [8,9]. The myomesin gene family contains three members (MYOM1, MYOM2, and MYOM3) and encodes myosin proteins that play crucial roles in heart and skeletal muscle development [10,11]. Myomesin is a molecular spring whose elasticity protects the stability of myosins, similar to actin [12]. Although the three proteins encoded by the Myomesin gene family have a similar structure, their expression in muscle tissue is significantly different [13]. Myomesin isoforms are associated with the contractile properties of different fiber types. For example, myomesin 1 is expressed in all transverse muscles, myomesin 2 is expressed in adult heart and fast fibers and myomesin 3 is expressed only in skeletal muscle intermediate fiber types [14].

Currently, there is no mammalian model of MYOM2, and only myomesin 3 has been studied in detail using zebrafish [15]. *Drosophila* is a classical model organism with a short life cycle and a relatively simple genetic structure characterized by a smaller number of genomic repeats than vertebrates [16]. About 75% of human disease genes have homologs in *Drosophila* [17,18]. *Drosophila* is the only experimental invertebrate model with a heart that is homologous to vertebrates in terms of cardiac development and function [19]. In addition, it has highly conserved biochemical and genetic pathways with mammals, and its cardiac decline is highly similar to that of the human heart. Furthermore, the mature genetic toolkit of *Drosophila* makes it suitable for studying MYOM2. dMnM is the homolog of MYOM2 in *Drosophila* and is expressed in *Drosophila* heart and skeletal muscle [6]. *Drosophila* can also be constructed to overexpress or knockout any gene in tissues in a time-specific manner, making it an excellent model for ageing studies. Muscle function can be easily determined by measuring the ability of *Drosophila* to fly and climb [20]. *Drosophila* has been widely used to study muscle development and disease. Skeletal muscle accounts for 75% of body mass in flying organisms, such as *Drosophila*, and flight muscles alone account for 65% of total body mass [21]. The IFM of the *Drosophila* thorax is the largest and the only fibrous muscle present in *Drosophila* [22]. Furthermore, the constituent proteins of the IFM show a high degree of homology with their vertebrate counterparts at the sequence, structural, and functional levels [23]. IFM mainly provides power for flight. Although studies have shown that MYOM2 serves as a major component of the myofibrillar myogenic fiber M-band and as a pivotal gene in myofibrillar gene interactions [6], the role of MYOM2 in cardiac function and IFM is unknown.

Oxidative stress is closely related to cardiomyopathy. Furthermore, oxidative damage to the myocardium may be due to low levels of antioxidant enzymes [24]. In addition, the effects of oxidative stress injury on myocardial toxicity are characterized by decreased antioxidant stress products, including glutathione peroxidase (GPX), catalase (CAT), and superoxide dismutase (SOD) [24]. Glycolipotoxicity increases the production of reactive oxygen species (ROS) by stimulating cardiomyocytes, thus enhancing the development of diabetic cardiomyopathy (DCM) [25,26,27,28]. Oxidative stress also promotes the development of alcoholic cardiomyopathy (ACM) [29]. Oxidative stress may also play a key role in hypertrophic cardiomyopathy (HCM) [30], where the development and progression of HCM depend on primary damage to myofascicles caused by mutations. However, the development and progression of HCM may also be associated with secondary alterations in the heart due to increased oxidative stress [31], leading to the exacerbation of myofascicular protein mutations. However, the relationship between dMnM knockdown in *Drosophila* cardiomyocytes and oxidative stress in the heart is unknown.

Regular exercise can reduce the incidence of cardiovascular disease [32,33]. Exercise may even lead to favorable cardiac remodeling [34,35] in mammals or *Drosophila*. In addition, exercise is a cost-effective way of preventing or improving certain heart diseases. For example, exercise attenuates HFD-induced cardiac fibrosis, reductions in the shortening fraction, and arrhythmias in rats and *Drosophila* [36,37,38]; improves cardiac dysfunction and lipid accumulation induced by Nmnat knockdown in *Drosophila* [39]; improves cardiac function and myocardial fibrosis in diabetic cardiomyopathy mice [40]; and prevents or ameliorates thioacetamide (TAA)-induced cardiac-dysfunctional pathological cardiac structural remodeling [41]. However, functional studies of MYOM2 in the heart and IFM are lacking, and the relationship between regular exercise and MYOM2 is unclear. Furthermore, the biological functions of regular exercise and MYOM2 and their role in cardiomyopathy are unknown. In this study, a *Drosophila* model was used to demonstrate the role of dMnM in cardiac function. Results showed that regular exercise could reverse the cardiac function defects caused by low dMnM expression. Regular exercise also reversed the reduction in climbing ability and shortening of lifespan caused by the low expression of dMnM in the IFM of *Drosophila*.

## 2. Materials and Methods

### 2.1. Drosophila Strains and Groupings

All *Drosophila* populations were fed on yeast–maltose–cornmeal–sucrose–agar standard food medium. The *Drosophila* were fed in clear glass tubes (Canghzhou four stars glass; Canghzhou; China)(20–30 flies in each tube) at a constant temperature in a humidity incubator CIMO, Shanghai, China()(25 °C, 50% humidity, 12 h day/night cycle). Wild-type *Drosophila melanogaster* was used: W1118 (3605, Bloomington *Drosophila* Stock Center). *Hand-Gal4*, 48396 (W1118; P{GMR88D05-Gal4}attP2, Bloomington *Drosophila* Stock Center). *Act88F-Gal4*,38461[w*; P{Act88F-GAL4.1.3}3, Bloomington *Drosophila* Stock Center]; for dMnM, RNAi GD: 65245 (y^1^ sc* v^1^ sev^21^; P{TRiP.HMC06031}attP2/TM3, Sb^1^, Bloomington *Drosophila* Stock Center).

The W1118 females were crossed with Hand-Gal4 and Act88F-Gal4 males, and F1 generation fledged females *W1118 > Hand-Gal4* and *W1118 > Act88F-Gal4* were collected within 12 h. *UAS-dMnM^RNAi^* females were crossed with *Hand-Gal4* and *Act88F-Gal4* males, and *UAS-dMnM^RNAi^* flies were collected within 12 h. *UAS-dMnM^RNAi^* flies were crossed with Hand-Gal4 and Act88F-Gal4 males, and *UAS-dMnM^RNAi^* flies were collected within 12 h. Females of F1 generation plumage *UAS-dMnM^RNAi^ > Hand-Gal4* and *UAS-dMnM^RNAi^ > Act88F-Gal4* were collected within 12 h. *UAS-dMnM^RNAi^* females were crossed with Hand-Gal4 and Act88F-Gal4 males, and females of F1 generation plumage were collected within 12 h and subjected to two weeks of regular movement (*UAS-dMnM^RNAi^ > Hand-Gal4 + E* and *UAS-dMnM^RNAi^ > Act88F-Gal4 + E*). All *Drosophila* were 21 day old females.

### 2.2. Exercise Programs

A locomotor device was designed to induce *Drosophila* to climb upward based on the natural negative ground-tending behavior of *Drosophila* [42]. Vials were loaded horizontally into a steel tube with 20 *Drosophila* per tube, then rotated around the horizontal axis at a gear-controlled axial speed, with each vial rotating along its long axis. Most *Drosophila* actively climbed upward during the rotation, and the few that could not climb actively walked on the inner walls of the vials. The vials were rotated at 24 s/revolution, and the exercise was conducted for 2.5 h per day [42] for two weeks with two days of rest.

### 2.3. Semi-Intact Drosophila Heart Preparation and Cardiac Function

First, 30 flies were fixed on Petri dishes after being anesthetized with FlyNap (Sangon Biotech, Shanghai, China) for 2–3 min. The head, thorax, cuticle, and all internal organs (except the heart) were quickly removed. Dissection was then performed using haemolymph containing 108 mM NaCl_2_, 5 mM KCl, 2 mM CaCl_2_, 8 mM MgCl_2_, 1 mM NaH_2_PO_4_, 4 mM NaHCO_3_, 15 mM 4-(2-hydroxyethyl)-1-piperazinebisulfonic acid, 10 mM sucrose, and 5 mM alginate at a pH of 7.1 and room temperature (24 °C) to expose the heart tube (to visualize the beating heart of the *Drosophila*) [43]. Oxygen was pumped at room temperature for 15 min. An EM-CCD 9300 high-speed camera (Hamamatsu; Shizuoka; Japan 100–140 fps/sec) taking high-speed digital videos of the beating heart at 130 fps/sec was used to record *Drosophila* heartbeats, while HCImage software (Hamamatsu; Shizuoka; Japan) was used to record recording ECG data. *A* semi-automatic optical heartbeat analysis (SOHA, provided by Ocorr and Bodmer) was used to analyze *Drosophila* cardiac function parameters. SOHA allows accurate quantification of heart rate (HR), cardiac cycle (HP), diastolic diameter (DD), systolic diameter (SD), systolic interval (SI), diastolic interval (DI), arrhythmia index (AI), fibrillation (FL), and shortening fraction (FS) for the assessment of *Drosophila* heart function parameters. In addition, cardiac M-patterns, such as qualitative recordings showing the temporal motion of the heart edges, were generated using an optical heartbeat analysis for further analysis of abnormal cardiac contractions [44].

### 2.4. qRT-qPCR

Total RNA was isolated from the heart tissues using TRIzol (Invitrogen, CA, USA) following the manufacturer’s protocol. The total RNA was reverse transcribed using Superscript III reverse transcriptase (Invitrogen, CA, USA) to generate cDNA, which was used as a template for quantitative real-time PCR. Real-time PCR was performed using the ABI7300 Real-time PCR Instrument (Takara; Beijing; China) with SYBR Green. The relative abundance of genes tested was calculated using the 2^−ΔΔCt^ method. The following primers were used: dMnM primer: F = 5′-CAGACCTGCGATTGCGGAAGTAG-3′; R = 5′-AGCGTGCGTCCTTGAACAAGTAC-3′. rp49 primer: F = 5′-CTAAGCTGTCGCACAAATGG-3′; R = 5′-AACTTCTTGAATCCGGTGGG-3′. upheld primer. F = 5′-AAGGATGCAGGCTTGGGAC-3′; R = 5′-GTGTCCACACGTCGCTCATA-3′. cat primer: F = 5′-TGCAAGTTCCCCAGTTGGAC-3′; R = 5′-CGATCTGCTCCACCTCAGCA-3′. sod2 primer: F = 5 ′-ATTTCGCAAACTGCAAGCCTGG-3′; R = 5′-TCTTCAGATCATCGCTGGGC-3′. phgpx primer: F = 5′-TCAGAACATTTCCCGCCAGG-3′; R = 5′-GTCGGGGCATATCGGTTGAT-3′.

### 2.5. Phalloidine

Semi-intact *Drosophila* hearts were prepared and confirmed to show rhythmic beating in oxygenated ADH [44]. ADH was quickly replaced with a relaxation buffer (ADH containing 10 mM EGTA). The hearts were fixed with 4% formaldehyde at room temperature for 20 min, then washed thrice using PBS at room temperature (10 min for each wash). The hearts were stained with ghost pen cyclic peptide (Phalloidin-iFluor 594) for 40 min and washed thrice with PBS at room temperature (10 min for each wash). Fluorescence staining images were obtained with a confocal laser scanning microscope (Carl Zeiss; Oberkochen; Germany).

### 2.6. ROS Staining

The flies were dissected to expose the cardiac canal as described by Alexander Lam et al. [45] and then stained for reactive oxygen ROS. The *Drosophila* were fixed after continuous exposure to CO_2_ for 1 min before dissection. The adult flies were dissected to maintain a semi-intact beating heart before staining. The fly was immersed in 1:1000 dihydroethidium (DHE, HY-D0079). PBS for 30 min after exposing the heart tube. The flies were rinsed thrice with PBS at room temperature (10 min for each wash). A Leica stereomicroscope (Leica; Wetzlar; Germany) was used to obtain images, which were then processed with Adobe Photoshop (Adobe, CA, USA).

### 2.7. Climbing and Lifespan Statistics

The *Drosophila* climbing apparatus consisted of an 18 cm long clear glass tube (Canghzhou four stars glass; Canghzhou; China) with a diameter of 2.8 cm and two 2 cm plugs (Canghzhou four stars glass; Canghzhou; China) at the ends of the glass to prevent *Drosophila* from climbing out. The *Drosophila* were gently shaken at the bottom and aggressively climbed upwards according to their own negative tropism [20]. The climbing was recorded using a video camera (Sony; Tokyo; Japan), and the fourth, fifth, and sixth images were intercepted at the end of the sixth second. The glass tube was divided into nine equal regions, which were scored as 1, 2, 3, 4, 5, 6, 7, 8, and 9 from bottom to top, and the number of *Drosophila* in each region was counted. The climbing index was calculated as follows: climbing index = total score/total number of flies.

Lifespan statistics of *Drosophila* were counted daily from the day of plumage until the last *Drosophila* died. The medium was changed every 24 h. The average lifespan and survival curve were used to characterize the lifespan. The sample contained about 200 individuals per group [46].

### 2.8. Statistical Analyses

Data analysis was conducted using SPSS version 22.0 software(SPSS, Chicago, IL, USA). An independent-samples *t*-test was used to assess the differences between groups. However, nonparametric tests were used when the variances were not equal. Data are expressed as means ± the standard error of the mean (SEM), α = 0.05, * *p* < 0.05, ** *p* < 0.01, *** *p* < 0.001.

## 3. Results

### 3.1. Drosophila Cardiomyocyte dMnM Knockdown Causes Cardiac Function Defects

Cardiac function in *Drosophila* is widely evaluated using M-mode ECG. In this study, heart rate (HR), arrhythmia index (AI), shortening fraction ((FS% = (DD − SD)/DD × 100%), systolic diameter (SD), diastolic diameter (DD), diastolic interval (DI), systolic interval (SI), cardiac cycle (HP), and fibrillation (FL) were also assessed using M-mode ECG.

The UAS-Hand-Gal4 system was used to determine the effect of dMnM knockdown in *Drosophila* cardiomyocytes on cardiac function. Compared with the *W1118 > Hand-Gal4* group, the heart rate, inter-systolic interval, and shortening fraction significantly decreased in the *dMnMRNAi > Hand-Gal4* group (Figure 1). In addition, cardiac cycle, diastolic interval, arrhythmia index, and fibrillation significantly increased in the *dMnMRNAi > Hand-Gal4* group. Meng Ding et al. [47] indicated that arrhythmias include heart rate, arrhythmia index, and fibrillation. In this study, dMnM knockdown decreased heart rate, increased the arrhythmia index, and increased fibrillation, indicating an increase in arrhythmias. Furthermore, dMnM knockdown increased diastolic time, decreased shortening fraction, and decreased cardiac pumping capacity. However, dMnM knockdown did not significantly change the diastolic diameter or systolic diameter, suggesting that the cardiomyocyte dMnM knockdown cannot induce changes in *Drosophila* heart diameter and systolic diameter, indicating diminished cardiac contractile performance. In conclusion, these results suggest that cardiomyocyte dMnM knockdown causes defects in cardiac function.

### 3.2. dMnM Knockdown in Drosophila Cardiomyocytes Causes Myofibril Destruction

The adult *Drosophila* melanogaster heart consists of a single layer of contractible monolayer cardiomyocytes that forms a simple linear tube.

The changes in myogenic fibers in the myocardium were examined to explore the changes in myogenic fibers associated with genetic mutations and regular motility. The F-actin was labeled with a ghost pencil loop peptide. The changes in the *Drosophila* heart tube were observed using confocal microscopy. Compared with the *W1118 > Hand-Gal4* group(Figure 2A), the *dMnM^RNAi^ > Hand-Gal4* group showed disruption of myogenic fibers(Figure 2B), suggesting that dMnM knockdown in cardiac myocytes severely damages myogenic fiber structure in *Drosophila* hearts.

### 3.3. dMnM Knockdown in Drosophila Cardiomyocytes Increases Cardiac ROS Production and Decreases mRNA Expression of dMnM, Upheld, sod2, Cat, and phgpx

Reactive oxygen species (ROS) production was assessed using fluorescent dye dihydroethidium (DHE). qRT-PCR was used to detect dMnM knockdown efficiency. Compared with the *W1118 > Hand-Gal4* group, mRNA expression of dMnM, upheld (cardiac troponin, an important marker of myocardial injury), and the antioxidant enzymes sod2, cat, and phgpx (homologs of SOD2, CAT, and GPX4) significantly decreased in the *dMnM^RNAi^ > Hand-Gal4* group.

DHE staining of *Drosophila* hearts was used to determine the relationship between low dMnM expression in *Drosophila* cardiomyocytes and oxidative stress in the heart in the *W1118 > Hand-Gal4* group and the *dMnM^RNAi^ > Hand-Gal4* group. Cardiomyocyte dMnM knockdown resulted in brighter cardiac DHE staining fluorescence, indicating that knockdown increases cardiac ROS production and oxidative stress (Figure 3A–C). In addition, qRT-PCR was used to assess isolated *Drosophila* hearts, including dMnM, upheld, sod2, cat, and phgpx. Compared with the *W1118 > Hand-Gal4* group, the mRNA expression levels of dMnM, upheld, sod2,cat and phgpx were significantly decreased in the *dMnM^RNAi^ > Hand-Gal4* group (Figure 3D–H). dMnM mRNA expression decreased by about 80%, indicating that the cardiomyocyte dMnM knockdown strain upheld is a key marker of myocardial injury. The decreased expression of upheld indicated myocardial injury, while the decreased expression of phgpx, sod2, and cat indicated that *Drosophila* cardiomyocyte dMnM knockdown reduced cardiac antioxidant capacity.

### 3.4. Effects of dMnM Knockdown in Cardiomyocytes and IFM on Climbing and Lifespan of Drosophila

dMnM was knocked down in *Drosophila* cardiomyocytes and indirect flight muscle to explore the effects of dMnM knockdown in cardiomyocytes and IFM on the climbing and lifespan of *Drosophila*. Compared with the *W1118 > Hand-Gal4* group, cardiomyocyte dMnM knockdown reduced the climbing ability of the *dMnM^RNAi^ > Hand-Gal4* group (Figure 4A). However, cardiomyocyte dMnM knockdown did not affect lifespan (Figure 4B,C). Compared with the *W1118 > Hand-Gal4* group, IFM dMnM knockdown significantly decreased the climbing ability and lifespan of the *dMnM^RNAi^ > Act88F-Gal4* group (Figure 4D–F). These results suggest that dMnM knockdown in cardiac myocytes affects the climbing ability of *Drosophila* and has less effect on lifespan, while IFM dMnM knockdown significantly affects the climbing ability and lifespan of *Drosophila*.

### 3.5. Regular Exercise Ameliorates Cardiac Function Defects Caused by dMnM Knockdown in Drosophila Cardiomyocytes

The flies underwent regular exercise for two weeks after cardiomyocyte dMnM knockdown. Compared with the *dMnM^RNAi^ > Hand-Gal4* group, cardiac cycle, arrhythmia index, fibrillation, and diastolic interval were significantly reduced in the *dMnM^RNAi^ > Hand-Gal4 + E* group (Figure 5B,F,H,I). However, heart reat and shortening fraction were significantly increased(Figure 5A,C). These results indicate that regular exercise can significantly improve and reverse the arrhythmia and diastolic dysfunction caused by dMnM knockdown in *Drosophila* cardiomyocytes. Regular exercise improved the shortening fraction of the heart and increased the pumping function of the heart based on the decreased systolic diameter and increased shortening fraction(Figure 5D). However, regular exercise did not significantly change diastolic diameter and systolic intervals (Figure 5E,G), suggesting that regular exercise does not alter the changes in cardiac diameter caused by dMnM knockdown in cardiomyocytes. In addition, regular exercise reduced fibrillation in the *dMnM^RNAi^ > Hand-Gal4 + E* group. In conclusion, these results suggest that regular exercise reverses dMnM knockdown-induced arrhythmias and diastolic dysfunction in *Drosophila* cardiomyocytes.

### 3.6. Regular Exercise Improves Myocardial Myogenic Fiber Destruction Caused by Cardiomyocyte dMnM Knockdown

The myocardial F-actins of the *dMnM^RNAi^ > Hand-Gal4* group and *dMnM^RNAi^ > Hand-Gal4 + E* group were subjected to fluorescence image observation to investigate the effect of regular exercise on cardiac myogenic fiber destruction caused by dMnM-specific knockdown of cardiomyocytes. The *dMnM^RNAi^ > Hand-Gal4 + E* group was also subjected to regular exercise for two weeks. Compared with the *dMnM^RNAi^ > Hand-Gal4* group(Figure 6A), the *dMnM^RNAi^ > Hand-Gal4 + E* group showed a tighter myofilament arrangement (Figure 6B), indicating that regular exercise protects against the myofibril destruction caused by dMnM knockdown in cardiac myocytes.

### 3.7. Regular Exercise Reduces ROS Production and Upregulates mRNA Expression of dMnM, Upheld, phgpx, sod2, and Cat in Cardiomyocytes after dMnM Knockdown

DHE staining of *Drosophila* hearts was conducted in the *dMnM^RNAi^ > Hand-Gal4* group and the *dMnM^RNAi^ > Hand-Gal4 + E* group to determine the effect of regular exercise on cardiac oxidative stress caused by dMnM knockdown in cardiomyocytes. Compared with the *dMnM^RNAi^ > Hand-Gal4* group, the DHE staining of hearts in the *dMnM^RNAi^ > Hand-Gal4 + E* group had darker fluorescence intensity, indicating that regular exercise reduced cardiac ROS production and cardiac oxidative stress (Figure 7A,B).

Proper exercise facilitates cardiac remodeling and improves the antioxidant capacity of the heart. However, the relationship between regular exercise and cardiac dMnM-specific knockdown resulting in reduced expression of dMnM, upheld, sod2, cat and phgpx is unknown. qRT-PCR assays showed that the mRNA levels of dMnM, upheld, sod2, cat and phgpx were significantly upregulated in *dMnM^RNAi^ > Hand-Gal4 + E* group compared with the *dMnM^RNAi^ > Hand-Gal4* group (Figure 7D–H). Regular exercise significantly alleviated the effect of dMnM knockdown in Drosophila, indicating that exercise rescued or ameliorated the cardiac knockdown damage to the heart. The upregulation of sod2, cat and phgpx after regular exercise increased the antioxidant level of the heart.

Proper exercise facilitates cardiac remodeling and improves the antioxidant capacity of the heart. However, the relationship between regular exercise and cardiac dMnM-specific knockdown resulting in reduced expression of dMnM, upheld, phgpx, sod2, and cat is unknown. qRT-PCR assays showed that the mRNA levels of dMnM, upheld, phgpx, sod2, and cat were significantly upregulated in the *dMnM^RNAi^ > Hand-Gal4 + E* group compared to the *dMnM^RNAi^ > Hand-Gal4* group (Figure 7D–H). Regular exercise significantly alleviated the effect of dMnM knockdown in *Drosophila*, indicating that exercise reversed or ameliorated the cardiac knockdown damage to the heart. The upregulation of phgpx, sod2, and cat after regular exercise increased the antioxidant level in the heart.

### 3.8. Effects of Regular Exercise on Climbing and Lifespan after dMnM Knockdown in Cardiac Myocytes and IFM of Drosophila

Regular exercise improved the climbing ability in the *dMnM^RNAi^ > Hand-Gal4 + E* group more significantly than in the *dMnM^RNAi^ > Hand-Gal4* group after dMnM knockdown in cardiac myocytes (Figure 8A). However, regular exercise did not significantly affect lifespan in the *dMnM^RNAi^ > Hand-Gal4 + E* group compared to the *dMnM^RNAi^ > Hand-Gal4* group (Figure 8B,C). In contrast, regular exercise significantly increased the climbing ability and lifespan of *Drosophila* after dMnM knockdown in cardiomyocytes and IFM compared to the *dMnM^RNAi^ > Act88F-Gal4* group (Figure 8D–F).

## 4. Discussion

The effect of dMnM knockdown on cardiac functional defects, such as cardiac arrhythmias and diastolic dysfunction, is unknown. Furthermore, the relationship between exercise, dMnM expression, and cardiac function is unclear. This study aimed to illustrate the effects of low dMnM expression on cardiac function and of regular exercise on dMnM in cardiomyocytes using a *Drosophila* model.

Cardiac pumping function is mainly determined by systolic and diastolic functions, the percentage reduction in wall diameter during the systole, and the shortening fraction (change in diameter); thus, it can be used to estimate the contractility of the *Drosophila* heart [48]. In this study, dMnM knockdown in *Drosophila* cardiomyocytes increased the systolic diameter of the heart. This decreased the shortening fraction, thus reducing the heart pumping capacity. A previous study showed that a moderate reduction in *Drosophila* dMnM results in cardiac dilation, while a complete loss or strong knockdown causes restriction [6]. In this study, cardiomyocyte dMnM knockdown did not significantly change diastolic diameter, indicating that cardiomyocyte dMnM knockdown cannot cause myocardial dilation, possibly due to the *Drosophila* genetic background and varying diets. Interestingly, dMnM knockdown prolonged diastolic interval and increased arrhythmias. The increased diastolic interval indicates diastolic dysfunction [49], characterized by impaired relaxation, decreased dilatability, and increased myocardial stiffness, which may be caused by excessive actin interactions [50,51]. However, this suggests that the role of dMnM is not limited to cardiac dilation. Meng Ding et al. [47] suggested that arrhythmias include heart rate, the arrhythmia index, and fibrillation. In this study, dMnM knockdown decreased heart rate, increased the arrhythmia index, and increased fibrillation, thus increasing arrhythmias. In addition, prolongation of the diastolic interval decreases cardiac contractility. These results suggest that dMnM plays an essential role in the heart. Cardiac function defects caused by dMnM knockdown in cardiomyocytes may be due to decreased cardiac antioxidant capacity. In addition, cardiac troponin T (cTnT) is a sensitive indicator of myocardial injury [52]. In this study, *Drosophila* cTnT may have been sensitive to low dMnM expression. Therefore, cTnT mutations can reduce the contractile performance of cardiomyocytes or even affect the whole heart when dMnM under-expression causes flocculation of myocardial contractile mechanisms [53]. Upheld is a cTnT homolog in *Drosophila*. In this study, dMnM knockdown in cardiomyocytes decreased mRNA expression of upheld, suggesting that cardiomyocyte dMnM knockdown may impair the contractile function of the myocardium.

Regular physical activity can reduce the risk of cardiovascular diseases [54,55,56]. In this study, regular exercise reduced the diastolic interval, heart rate, arrhythmia index, and fibrillation, suggesting that regular exercise can significantly protect against the development of cardiac diastolic defects and arrhythmias triggered by dMnM knockdown. In addition, regular exercise reduced the systolic diameter and increased the shortening fraction, suggesting that exercise improves cardiac contractility and pumping capacity. In conclusion, these results show that two-week regular exercise can protect *Drosophila* cardiomyocytes against diastolic defects and arrhythmias caused by dMnM knockdown.

Oxidative stress in the excessive production of ROS is relative to antioxidant levels. Reactive oxygen species are oxy-chemicals with high reactivity [57]. ROS have deleterious effects on cardiovascular diseases, such as heart failure, cardiomyopathy, and coronary artery disease [58]. Excess ROS cause cellular dysfunction, protein and lipid peroxidation, and DNA damage and can lead to irreversible cellular damage and death. Intracellular ROS are mainly scavenged by antioxidant enzymatic substances, including superoxide dismutase (SOD2), catalase (CAT), and glutathione peroxidase (GPX). Moreover, antioxidant enzymatic substances are also the most important intracellular antioxidants in humans [59]. The three antioxidant enzymatic substances (SOD2, CAT, and GPX) are homologs of sod2, cat, and phgpx in *Drosophila*, respectively. Exercise promotes health through the maintenance of the biological adaptation of mitochondrial function to oxidative stress. Furthermore, exercise intervention has effects on antioxidants, such as SOD2, CAT, and GPX. For example, a four-week exercise intervention can promote SOD2 expression in a healthy population (SOD2 is positively correlated with exercise habit) [60]. Furthermore, glutathione peroxidase 4 (GPX4) activated by nuclear factor erythroid 2-related factor 2 (Nrf2) has a protective effect on myocardial injury in mice on a high-fat diet during aerobic exercise [61]. In this study, cardiomyocyte dMnM knockdown decreased sod2, cat, and phgpx expressions, suggesting that cardiomyocyte dMnM knockdown reduces cardiac antioxidant capacity. However, regular exercise significantly upregulated the mRNA expression of sod2, cat, and phgpx compared to the knockdown group, suggesting that regular exercise improves the antioxidant capacity of the heart by upregulating the expression of antioxidant enzymes.

French et al. indicated that exercise-induced changes in key antioxidant enzymes play an essential role in exercise-induced cardioprotection [62]. Herein, the DHE cardiac ROS fluorometry showed that the knockdown group had higher brightness after dMnM knockdown in cardiomyocytes than the exercise group. The exercise group also had reduced ROS production compared to the knockdown group. These results confirm that regular exercise can reduce oxidative stress caused by dMnM knockdown in cardiomyocytes.

Furthermore, the knockdown group showed disrupted F-actin myofilament arrangement and altered cardiac morphology compared to the paired group after staining cardiomyocyte myogenic fibers with ghost pencil cyclic peptide. However, regular exercise made the F-actin myofilament arrangement tighter and more complete, suggesting that regular exercise can reverse the structural disruption of myofibrils caused by dMnM knockdown. Importantly, dMnM knockdown in cardiac myocytes only reduced the climbing ability of *Drosophila*, while dMnM knockdown in IFM decreased both the climbing ability and lifespan of *Drosophila*. However, regular exercise significantly improved the climbing ability and lifespan decreased by dMnM knockdown in cardiac myocytes and IFM.

In summary, dMnM knockdown in *Drosophila* cardiomyocytes increases cardiac arrhythmias, increases diastolic dysfunction, and reduces the shortening fraction, possibly due to increased cardiac oxidative stress. In addition, dMnM knockdown in IFM reduces the climbing ability and lifespan of *Drosophila*. However, regular exercise can upregulate cardiomyocyte dMnM expression levels and alleviate the cardiac dysfunction caused by cardiomyocyte dMnM knockdown by increasing cardiac antioxidant capacity. Regular exercise can also improve the reduced climbing ability and shortened lifespan of *Drosophila* caused by IFM dMnM knockdown. In conclusion, this study demonstrates the protective effect of regular exercise on cardiac function in cardiac myocytes.

## 5. Conclusions

*Drosophila* cardiomyocyte dMnM knockdown leads to cardiac functional defects, while IFM dMnM knockdown affects climbing and lifespan. Regular exercise can upregulate cardiomyocyte dMnM expression levels and ameliorate cardiac functional defects caused by *Drosophila* cardiomyocyte dMnM knockdown by increasing cardiac antioxidant capacity. In addition, regular exercise can reverse the shortening of lifespan caused by IFM dMnM knockdown.

## Figures and Tables

**Figure 1 ijerph-19-16554-f001:**
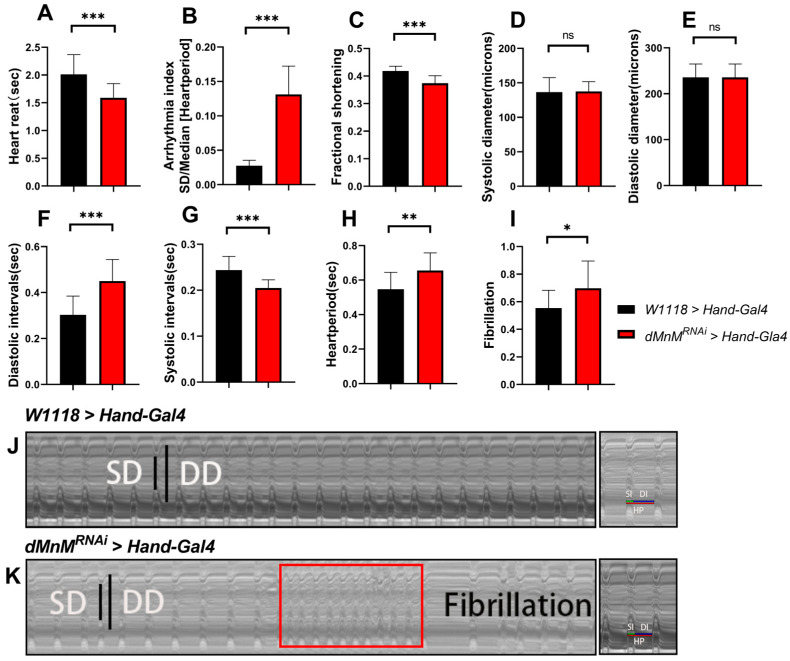
Effects of dMnM knockdown in cardiomyocytes on cardiac function. (**A**–**H**) M HR, AI, FS, SD, DD, DI, SI, HP, and FL in the *W1118 > Hand-Gal4* group and the *dMnM^RNAi^ > Hand-Gal4* group; all *Drosophila* females, n = 30. (**I**) *Drosophila* fibrillation level in *Drosophila*. (**J**,**K**) ECG pattern. Long and short black lines represent diastolic diameter and systolic diameter, respectively. The red square shows fibrillation. All M-mode ECGs were intercepted at 10 s. An independent-samples *t*-test was used to evaluate the differences between the two groups, * *p* < 0.05, ** *p* < 0.01, *** *p* < 0.001, ns is not statistically significant.

**Figure 2 ijerph-19-16554-f002:**
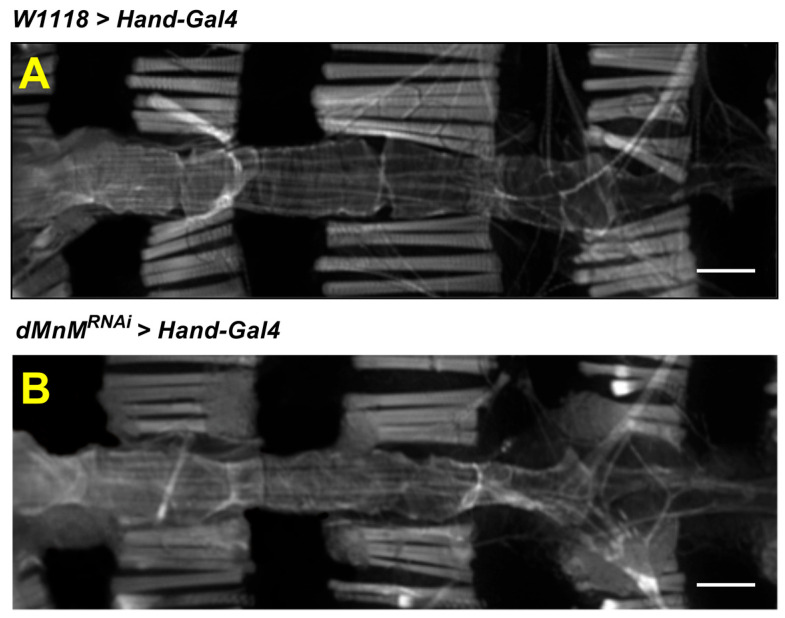
Effect of dMnM knockdown in cardiomyocytes on myofibrillar F-actin in cardiac myocytes. (**A**) Fluorescence images of the heart in the *W1118 > Hand-Gal4* group; (**B**) fluorescence images of the heart in the *dMnM^RNAi^ > Hand-Gal4* group, n = 5. A confocal microscope was used to observe the *Drosophila* heart tube with ghost pen cyclic peptide-labeled F-actin, scale bar = 100 μm.

**Figure 3 ijerph-19-16554-f003:**
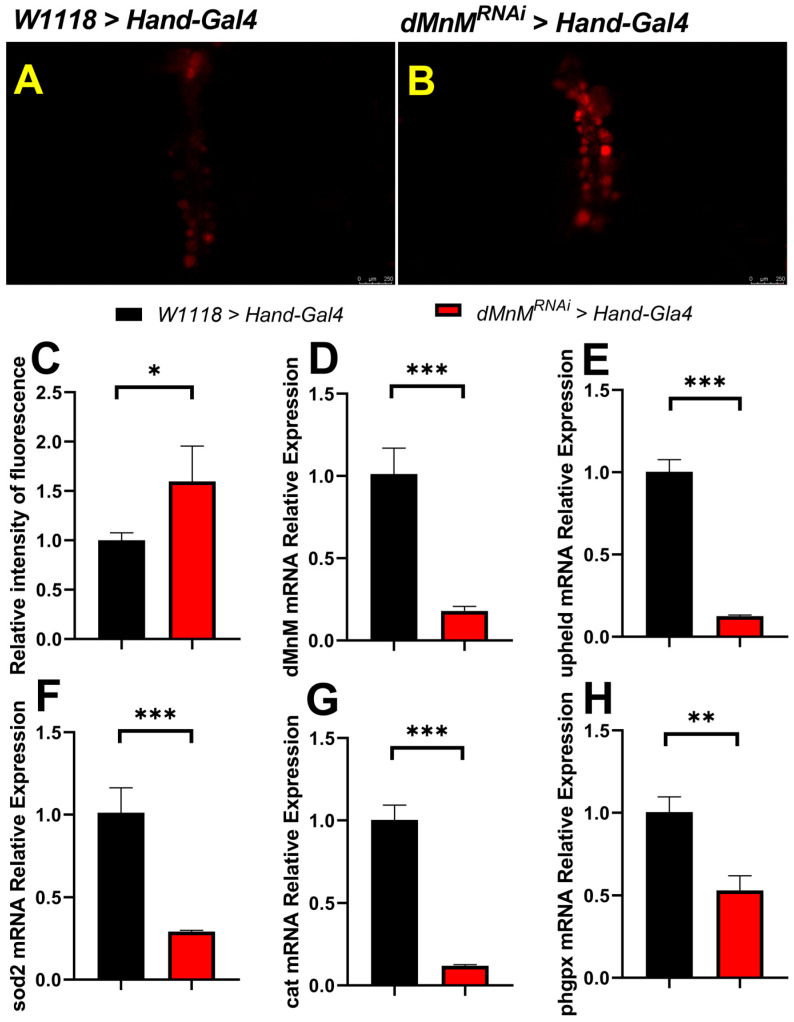
Effects of dMnM knockdown in cardiac myocytes on oxidative stress in the heart and mRNA expression of dMnM, upheld, sod2, cat, and phgpx. (**A**,**B**) Fluorescence images of cardiac DHE staining in the *W1118 > Hand-Gal4* and *dMnM^RNAi^ > Hand-Gal4* groups, respectively, n = 5. (**C**) Relative fluorescence intensity of the heart. (**D**–**H**) Relative expression levels of dMnM, upheld, sod2, cat, and phgpx mRNA in *Drosophila* cardiomyocytes, respectively, n = 30. T-test for differences between the two groups using independent samples, * *p* < 0.05, ** *p* < 0.01, *** *p* < 0.001.

**Figure 4 ijerph-19-16554-f004:**
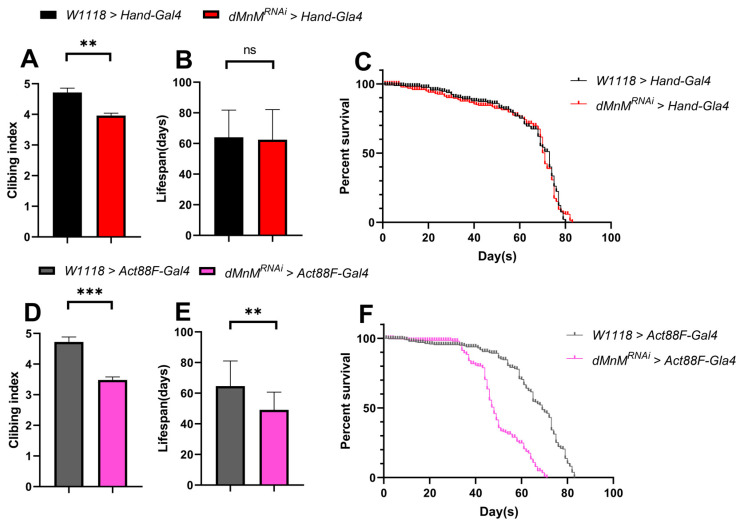
Effects of dMnM knockdown in cardiomyocytes and IFM on climbing and lifespan of *Drosophila*. (**A**) Climbing results for the *W1118 > Hand-Gal4* group; (**B**,**C**) lifespan and survival curves for the *W1118 > Hand-Gal4* group and *dMnM^RNAi^ > Hand-Gal4* group, respectively; (**D**) climbing results for the *W1118 > Act88F-Gal4* and *dMnM^RNAi^ > Act88F-Gal4* group; (**E**,**F**) lifespan and survival curves for *dMnM^RNAi^ > Hand-Gal4* group, respectively. Climbing sample n = 100; lifespan sample n = 200. The difference between the two groups was tested using an independent-samples *t*-test, ** *p* < 0.01, *** *p* < 0.001, ns is not statistically significant.

**Figure 5 ijerph-19-16554-f005:**
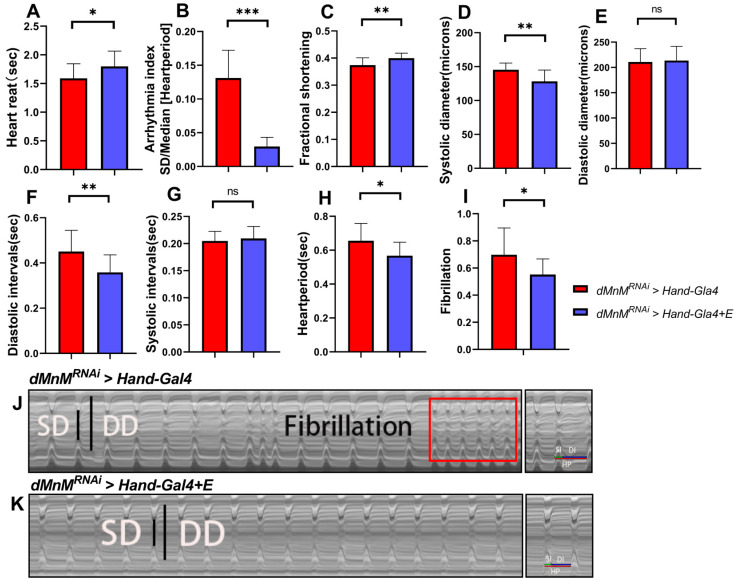
Effect of regular exercise on dMnM-specific knockdown of cardiac function in *Drosophila* cardiomyocytes. (**A**–**H**) HR, AI, FS, SD, DD, DI, SI, HP, and FL in the *dMnM^RNAi^ > Hand-Gal4* group and the *dMnM^RNAi^ > Hand-Gal4 + E* group, n= 30. (**I**) Fibrillation levels in Drosophila. (**J**,**K**) Drosophila M-mode ECG pattern. Long and short black lines represent diastolic diameter and systolic diameter, respectively. The red square shows fibrillation. Rectangular area represents fibrillation. All M-mode ECGs were intercepted for 10 s. An independent-samples *t*-test and nonparametric test were used to assess the differences between the two groups, * *p* < 0.05, ** *p* < 0.01, *** *p* < 0.001, ns is not statistically significant.

**Figure 6 ijerph-19-16554-f006:**
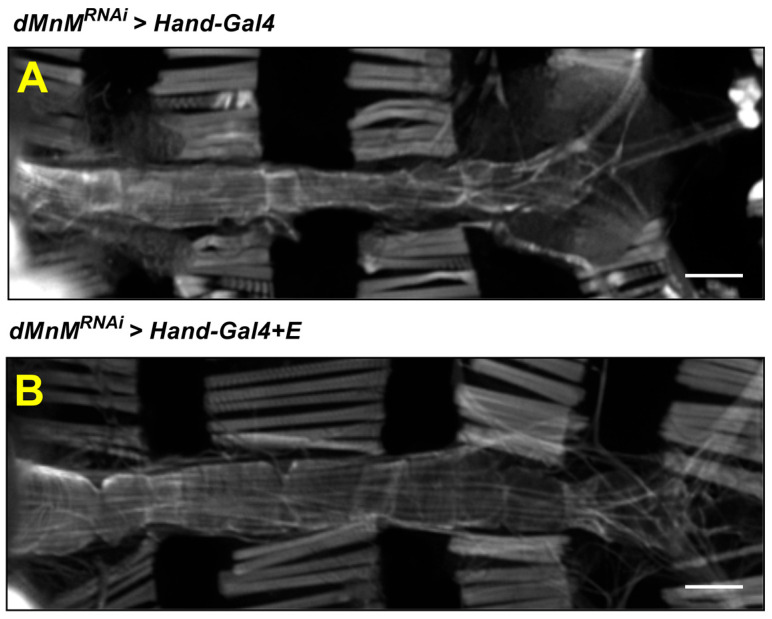
Effect of regular exercise on F-actin in myogenic fibers of cardiac myocytes after dMnM knockdown. (**A**) Fluorescence images of the heart in the *dMnM^RNAi^ > Hand-Gal4* group. (**B**) Fluorescence images of the heart in the *dMnM^RNAi^ > Hand-Gal4 + E* group, n = 5. F-actin was labeled with ghostwritten cyclic peptide. Confocal microscopy was used to observe the changes in the *Drosophila* heart tube.

**Figure 7 ijerph-19-16554-f007:**
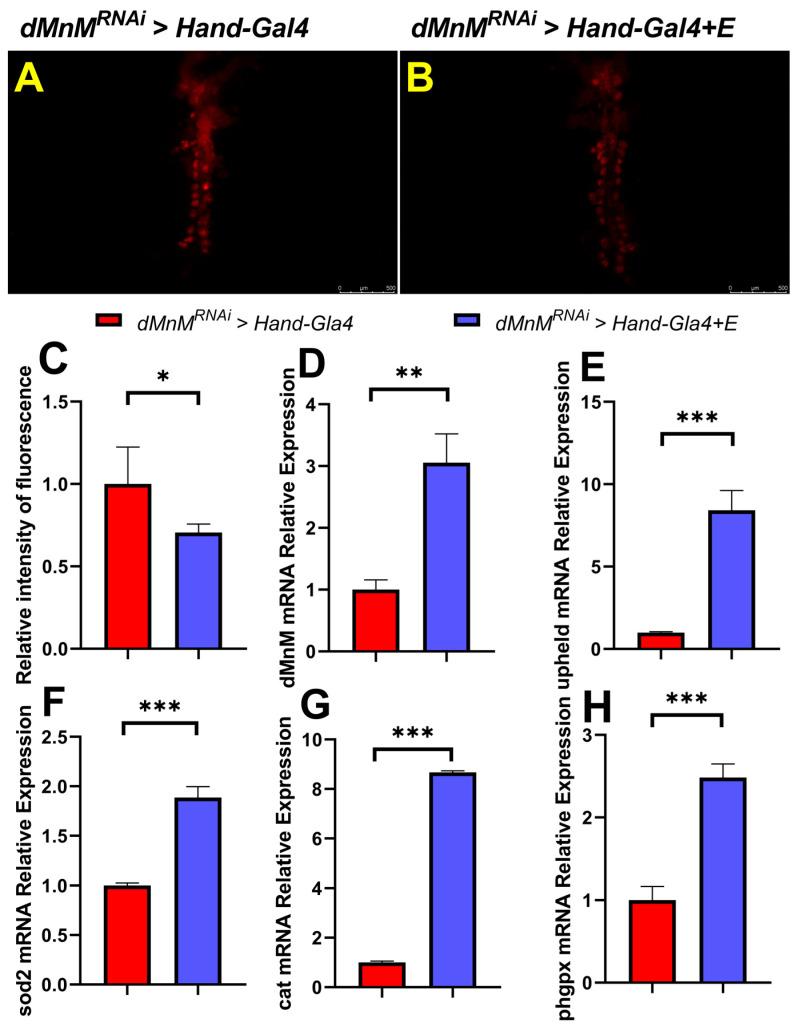
Effects of regular exercise on ROS produced by specific knockdown of cardiac dMnM and mRNA expression levels of dMnM, upheld, phgpx, sod2, and cat. (**A**,**B**) DHE-stained images of hearts in the *dMnM^RNAi^ > Hand-Gal4* group and *dMnM^RNAi^ > Hand-Gal4 + E* group, respectively; n = 5. (**C**) Relative fluorescence intensity of the heart. (**D**–**H**) Relative expression levels of dMnM, upheld, sod2, cat, and phgpx mRNA; n = 30. An independent-samples *t*-test was used to assess differences between the two groups, * *p* < 0.05, ** *p* < 0.01, *** *p* < 0.001.

**Figure 8 ijerph-19-16554-f008:**
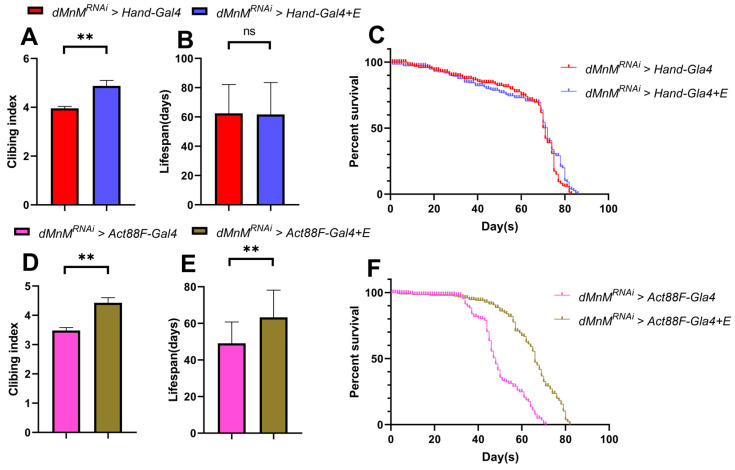
Effects of regular exercise on climbing and lifespan of Drosophila melanogaster after dMnM knockdown in cardiac myocytes and IFM. (**A**) Climbing results for the *dMnM^RNAi^ > Hand-Gal4* group. (**B**,**C**) Lifespan and survival curves for the *dMnM^RNAi^ > Hand-Gal4* group and *dMnM^RNAi^ > Hand-Gal4 + E* group, respectively. (**D**) Climbing results for the *dMnM^RNAi^ > Act88F-Gal4* and *dMnM^RNAi^ > Act88F-Gal4 + E* group. (**E**,**F**) Lifespan and survival curves of the *dMnM^RNAi^ > Hand-Gal4* and *dMnM^RNAi^ > Act88F-Gal4 + E* groups, respectively, Climbing sample n = 100; lifespan sample n = 200. An independent-samples *t*-test was used to assess the difference between the two groups, ** *p* < 0.01, ns is not statistically significant.

## Data Availability

The raw data supporting the conclusions of this article will be made available by the authors without undue reservation.

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
