# Peer review of "Regular Exercise Rescues Heart Function Defects and Shortens the Lifespan of Drosophila Caused by dMnM Downregulation"

_ijerph, 2022, doi:10.3390/ijerph192416554_

Round 1
Reviewer 1 Report
Dear authors
Drosophila cardiac function parameters were analyzed by semi-automatic op- 151 tical heartbeat analysis (SOHA, kindly presented by Ocorr and Bodmer), which allows 152 accurate quantification of heart rate (HR), cardiac cycle (HP), diastolic diameter (DD), sys- 153 tolic diameter (SD), systolic interval (SI), diastolic interval (DI), arrhythmia index (AI), 154 fibrillation (FL) and shortening fraction (FS) for the assessment of Drosophila heart funcion parameters. And then, in the figure 1, M mode detection was announced. Please explain precise method of cardiac function evaluation. Furthermore, not only results but also real example of function evaluation was needed.
Author Response
Dear Reviewer:
We sincerely thank the reviewer for thoroughly examining our manuscript and providing very helpful comments to guide our revision. And we have tried our best to revise the manuscript according to your kind and constructive on comments and suggestions. The responses to the comments are given below.
1 Drosophila cardiac function parameters were analyzed by semi-automatic op- 151 tical heartbeat analysis (SOHA, kindly presented by Ocorr and Bodmer), which allows 152 accurate quantification of heart rate (HR), cardiac cycle (HP), diastolic diameter (DD), sys- 153 tolic diameter (SD), systolic interval (SI), diastolic interval (DI), arrhythmia index (AI), 154 fibrillation (FL) and shortening fraction (FS) for the assessment of Drosophila heart funcion parameters. And then, in the figure 1, M mode detection was announced. Please explain precise method of cardiac function evaluation. Furthermore, not only results but also real example of function evaluation was needed.
Response: Thank you very much for your valuable suggestions! SOHA is a software that can accurately quantify heart rate (HR), cardiac cycle (HP), diastolic diameter (DD), systolic diameter (SD), systolic interval (SI), diastolic interval (DI), arrhythmia index (AI), fibrillation (FL) and shortening fraction (FS) for the assessment of cardiac function parameters in Drosophila.
SOHA is a well-established software that accurately quantifies heart rate (HR), cardiac cycle (HP), diastolic diameter (DD), systolic diameter (SD), systolic interval (SI), diastolic interval (DI), arrhythmia index (AI), fibrillation (FL) and shortening fraction (FS) for assessing cardiac function parameters in Drosophila. For the cardiac function test, a 30-second video of the ECG is taken and then this video is "dotted" on the SOHA software, with two red dots on the edge of the heart at the end of systole, which is the systolic diameter (Figure 1); similarly, two red dots on the edge of the heart at the end of diastole, which is the diastolic diameter (Figure 2). One cardiac cycle is from the beginning of contraction to the end of diastole (Figure 3), and one cardiac cycle is between the two green lines. In addition, these specific results such as HR, HP, DD, SD, SI, DI, AI, and FL can be directly derived by SOHA, except for FS (FS=(DD-SD)/DD). In addition, if there is a defect in cardiac function, such as an arrhythmia, the heartbeat will be very irregularly beating and the arrhythmia index and fibrillation will be increased.
Thanks again for your constructive suggestions.
Figures 1,2 and 3 are detailed in document 1

Reviewer 2 Report
General comments
The study shows that knockdown of dMnM in Drosophila cardiomyocytes resulted in cardiac diastolic defects, arrhythmias, and increased cardiac oxidative stress. Knockdown of Drosophila dMnM in indirect flight muscle(IFM) resulted in reduced exercise climbing ability and shortened lifespan. Authors found that regular exercise significantly rescued dysfunctions caused by dMnM knockdown.
The study appears to have appropriate methodology. The data are rather clearly presented, but in Figure 5 there in no information about the number of analyzed individuals. There is also an inaccuracy in the information about number of analyzed individuals. Whereas in Material and Method section (point 2.3 and 2.4) authors write about 30 flies in a group, in Figure 3 there is N=60 considering mRNA analyses. And the next question is arising: for mRNA experiment 30 hearts were pulled, so this way it is one mRNA and cDNA sample, so in Figure 3 and 7 are there 2 samples (60/30 = 2) in each analyzed group? Authors should explain it. Generally, I would recommend to define how many flies were used for a given experiment.
In Results section threee last points must be proved by authors:
· Where are results described in point 3.5. (Regular exercise ameliorates cardiac function defects caused by dMnM knockdown in 308 Drosophila cardiomyocytes)?
· Results described in point 3.6. (Regular exercise improves cardiomyocyte dMnM knockdown leading to myocardial myogenic fiber destruction and reduction of F-Actin) seem to presented on Fig. 6 not Fig. 5, please check it. There is no statystical analysis of these results!
· point 3.8 - the text is missing!
DHE staining is shown only on pictures (Fig. 3 and Fig. 7), there is no statistical analysis of these results, so statement ’ knockdown increased cardiac ROS production and increased cardiac oxidative stress’ is not scientifically based. I will recommend to do this analysis in both cases (Fig. 3 and 7) to show the oxidative stress, otherwise it is just a wishful thinking.
The paper in some fragments is well written, but in other not. Authors should read the manuscript carefully, especially Discussion section, and make some stylistic amendments. I would recommend to do a detailed language check to correct a structure of some sentences
Abstract
Line 12 – please explain the following abbveration: MYOM2.
Line 14 – please explain the following abbveration: dMnM.
Line 21 – 23 – this sentence is not clear, please rewrite it.
Introduction
Line 34 - 38 - this sentence is not clear, please rewrite it.
Line 89 – 92 - this sentence is completely unclear, please rewrite it.
Materials and Methods
I would recommend to define how many flies were used for a given experiment.
Point 2.4. qRT-qPCR – please explain all the abbverations for gene names.
Results
Line 212 – heart reat instead of heartrate.
Line 262 – 264 - please rewrite the sentence, it is unclear.
Line 265 – 270 - please rewrite the sentence, it is very unclear.
Line 265 - 269 - DHE staining is shown only on pictures (Fig. 3), there is no statistical analysis of this result, so statement ’ knockdown increased cardiac ROS production and increased cardiac oxidative stress’ is not scientifically based.
Line 270 -271 please rewrite the sentence, it is unclear.
Line 289 – 291 - please rewrite the sentence, it needs a stylistic amendment.
Line 308 – 323 Where are results described in point 3.5. (Regular exercise ameliorates cardiac function defects caused by dMnM knockdown in 308 Drosophila cardiomyocytes)?
Line 331 – 340 Results described in point 3.6. (Regular exercise improves cardiomyocyte dMnM knockdown leading to myocardial myogenic fiber destruction and reduction of F-Actin) seem to presented on Fig. 6 not Fig. 5, please check it.
There is no statystical analysis of these results!
Line 377 – point 3.8 - the text is missing!
Discussion
Line 390 – 392 Sentence ‘To address this question, we used a Drosophila model to elucidate the effects of dMnM in cardiac function and regular exercise and the upregulation of dMnM expression in cardiomyocytes by regular exercise.’ must be rewritten – make it clear to the reader.
Line 404 – 406 sentence ‘dMnM knockdown allows for increased diastolic defects and arrhythmias due to prolonged cardiac diastolic intervals, and the increased diastolic interval reminds us of diastolic dysfunction must be rewritten – make it clear to the reader.
Line 413 – 414 the sentence ‘This series of results suggests that dMnM plays an essential role in the heart.’ needs a stylistic amendment.
Line 414 – 417 – the sentence ‘We speculate that the cause of cardiac function defects due to low dMnM knockdown in cardiomyocytes may be related to the downregulation of mRNA expression of dMnM in cardiomyocytes and the reduced anti-oxidant capacity of the heart’ is not clear, please rewrite it.
Line 421 – please rewrite this sentence to make it more clear: ‘upheld that CTnT is homologous in Drosophila’
Line 435 – sentence ‘Oxidative stress, defined as the excessive production of ROS relative to antioxidant levels, reactive oxygen species are oxy-chemicals with high reactivity’ needs a stylistic amendment.
Line 461 – 469 – As there is no statistical analysis of DHE staining authors cannot draw described conclusions.
Line 470 – 476 - There is no statystical analysis of discussed results! Authors cannot draw described conclusions.
Author Response
Dear Reviewer :
We sincerely thank the reviewer for thoroughly examining our manuscript and providing very helpful comments to guide our revision. And we have tried our best to revise the manuscript according to your kind and constructive on comments and suggestions. The responses to the comments are given below.
1 The study appears to have appropriate methodology. The data are rather clearly presented, but in Figure 5 there in no information about the number of analyzed individuals. There is also an inaccuracy in the information about number of analyzed individuals. Whereas in Material and Method section (point 2.3 and 2.4) authors write about 30 flies in a group, in Figure 3 there is N=60 considering mRNA analyses. And the next question is arising: for mRNA experiment 30 hearts were pulled, so this way it is one mRNA and cDNA sample, so in Figure 3 and 7 are there 2 samples (60/30 = 2) in each analyzed group? Authors should explain it. Generally, I would recommend to define how many flies were used for a given experiment.
Response: Thank you very much for your valuable comments, and we apologize for the sample size, which was incorrectly stated due to the oversight of our writing students. In fact, for both cardiac function analysis and mRNA experiments, our sample size is 30 fruit flies. We have also made changes in the manuscript based on your suggestions.
2 In Results section threee last points must be proved by authors:
Where are results described in point 3.5. (Regular exercise ameliorates cardiac function defects caused by dMnM knockdown in 308 Drosophila cardiomyocytes)?
Response: Thank you very much for the reminder, we have added descriptive results to the manuscript: In conclusion, our results suggest that regular exercise rescues dMnM knockdown-induced arrhythmias and diastolic dysfunction in Drosophila cardiomyocytes.
3 Results described in point 3.6. (Regular exercise improves cardiomyocyte dMnM knockdown leading to myocardial myogenic fiber destruction and reduction of F-Actin) seem to presented on Fig. 6 not Fig. 5, please check it. There is no statystical analysis of these results!
Response: We agree with the viewpoint of reviewer. We apologize for the incorrect results described in points 3.2 and 3.6, as we did not perform a statistical analysis, we only elaborated that regular exercise protected cardiomyocytes from dMnM knockdown resulting in myocardial myogenic fiber destruction. We have completed the changes in the manuscript based on your suggestions.
4 point 3.8 - the text is missing!
Response: We are sorry for the omissions due to our writing mistakes. We have filled in the text of 3.8 in the manuscript. Regular exercise significantly improved the climbing ability in the dMnMRNAi>Hand-Gal4+E group than in the dMnMRNAi>Hand-Gal4 group after dMnM knockdown in cardiac myocyte (Figure 8A). However, regular exercise did not significantly affect lifespan in dMnMRNAi>Hand-Gal4+E group compared with the dMnMRNAi>Hand-Gal4 group (Figure 8B and 8C). In contrast, regular exercise significantly increased the climbing ability and lifespan of Drosophila after dMnM knockdown in cardiomyocytes and IFM compared with the dMnMRNAi>Act88F-Gal4 group (Figure 8D,8E and 8F).
5 DHE staining is shown only on pictures (Fig. 3 and Fig. 7), there is no statistical analysis of these results, so statement ’ knockdown increased cardiac ROS production and increased cardiac oxidative stress’ is not scientifically based. I will recommend to do this analysis in both cases (Fig. 3 and 7) to show the oxidative stress, otherwise it is just a wishful thinking.
Response: Thank you very much for your valuable comments. We further analyzed the DHE staining in Figure 3 and Figure 7 and performed a quantitative analysis. We have made changes in the manuscript. Thank you again for your valuable suggestions to us.
6 The paper in some fragments is well written, but in other not. Authors should read the manuscript carefully, especially Discussion section, and make some stylistic amendments. I would recommend to do a detailed language check to correct a structure of some sentences.
Response: Thank you very much for your offer! As you said, there are many problems in the manuscript, we have checked the language problems and touched up the whole text.
7 Abstract
Line 12 – please explain the following abbveration: MYOM2.
Line 14 – please explain the following abbveration: dMnM.
Line 21 – 23 – this sentence is not clear, please rewrite it.
Response: Thank you very much for your valuable comments.
Line 12: MYOM2(myomesin 2) is a major component of the myofibrillar M-band of the sarcomere, and a hub gene within interactions of sarcomere genes.
Line 14: dMnM is the homolog of MYOM2 in Drosophila.
Line 21 – 23: In the manuscript: “in conclusion, our evidence suggests that dMnM has a greater impact on cardiac function in the heart, where low expression triggers cardiac functional defects, and in IFM is more sensitive to lifespan, and knockdown of dMnM in IFM leads to reduced exercise climbing ability and shortened lifespan”. We have modified it to “In conclusion our evidence suggests that Drosophila cardiomyocyte dMnM knockdown leads to cardiac functional defects, while indirect flight muscle dMnM knockdown affects climbing and lifespan”.
8 Introduction
Line 34 - 38 - this sentence is not clear, please rewrite it.
Line 89 – 92 - this sentence is completely unclear, please rewrite it.
Response: Thank you very much for your valuable comments.
Firstly, Line 34 – 38: “Recent studies have identified mutations in the myojunctional gene MYOM2 in patients with HCM[6], and interestingly, patients have no mutations in the 12 most common HCM disease genes such as MYH7, MYBPC3, etc., thus, MYOM2 may serve as a new candidate gene for HCM pathogenesis”. Because we know that there are 12 common genes that cause HCM pathogenicity in patients, such as MYH7, MYBPC3, TNNT, MYL2, etc. However, a recent study found mutations in MYOM2 gene in HCM patients, and interestingly, they did not find mutations in these 12 common pathogenic genes in these patients, therefore, MYOM2 could potentially be a new candidate gene for HCM pathogenesis.
So, We rewrite as: Although recent studies have identified mutations in the myojunction gene MYOM2 in HCM patients [6], these patients have no mutations in the 12 most common pathogenic genes, such as MYH7, MYBPC3, etc. Therefore, MYOM2 may serve as a new candidate gene for HCM pathogenesis.
We have already completed the changes in the manuscript, thank you again for your suggestions.
Secondly, Line 89 – 92 “However, it is not known whether the knockdown of dMnM in Drosophila cardiomyocytes is also due to increased oxidative stress due to the production of cardiac ROS.” We regret that due to our translation error, we would like to clarify that the development and progression of HCM may also be associated with secondary alterations in the heart due to increased oxidative stress, leading to the exacerbation of myofascicular protein mutations. However, the relationship between dMnM knockdown in Drosophila cardiomyocytes and oxidative stress in the heart is unknown.
9 Materials and Methods
I would recommend to define how many flies were used for a given experiment.
Point 2.4. qRT-qPCR – please explain all the abbverations for gene names.
Response: Thank you very much for your valuable comments.
Indeed, as you said, our sample size was not clear in the manuscript, and we used 30 Drosophila heart samples per group in the qRT-PCR and cardiac function assays, which we have revised in the manuscript based on your suggestion.
In qRT-PCR experiments we examined the mRNA expression of dMnM, upheld, sod2, cat, and phgpx. Among them, dMnM is the homolog of MYOM2 (myomesin 2 ) in Drosophila; upheld is the homolog of cardiac troponin (cTnT) in Drosophila; sod2 is the homolog of superoxide dismutase (SOD2) in Drosophila; cat is the homolog of catalase (CAT) in Drosophila; phgpx is the homolog of glutathione peroxidase ( GPX4) homolog in Drosophila.
10 Line 212 – heart reat instead of heartrate.
Line 262 – 264 - please rewrite the sentence, it is unclear.
Line 265 – 270 - please rewrite the sentence, it is very unclear.
Line 265 - 269 - DHE staining is shown only on pictures (Fig. 3), there is no statistical analysis of this result, so statement knockdown increased cardiac ROS production and increased cardiac oxidative stress’ is not scientifically based.
Response: Thank you very much for your valuable comments.
10 Line 212: We have completed the changes in the manuscript according to your requirements.
Line 262 – 264: We apologize for this because we wrote incorrectly, and we would like to express: DHE staining of Drosophila hearts was used to determine the relationship between low dMnM expression in Drosophila cardiomyocytes and oxidative stress in the heart in the W1118>Hand-Gal4 group and the dMnMRNAi>Hand-Gal4 group.
Line 265 – 270: Thank you very much for your valuable comments.
Sorry due to our translation error, we would like to show that: we performed qRT-PCR was used to assess isolated Drosophila hearts, including dMnM, upheld, sod2, cat, and phgpx. Compared with the W1118>Hand-Gal4 group, the mRNA expression levels of dMnM, upheld, phgpx, sod2, and cat were significantly decreased in the dMnMRNAi>Hand-Gal4 group (Figure 3C, D, E, F, and G). dMnM mRNA expression decreased by about 80%, indicating that the cardiomyocyte dMnM knockdown strain Upheld is a key marker of myocardial injury. The decreased expression of upheld indicated myocardial injury, while the decreased expression of phgpx, sod2, and cat indicated that Drosophila cardiomyocyte dMnM knockdown reduced cardiac antioxidant capacity.
Line 265 – 269:
We have performed statistical analysis of the fluorescence intensity of the cardiac fluorescence images as you requested, in a manuscript completed with modifications(N=5).
11 Line 270 -271 please rewrite the sentence, it is unclear.
Line 289 – 291 - please rewrite the sentence, it needs a stylistic amendment.
Line 308 – 323 Where are results described in point 3.5. (Regular exercise ameliorates cardiac function defects caused by dMnM knockdown in 308 Drosophila cardiomyocytes)?
Thank you very much for your valuable comments.
Line 270 -271:
Response: Thank you very much for your valuable comments.
We have modified: Compared with the W1118>Hand-Gal4 group, the mRNA expression levels of dMnM, upheld, phgpx, sod2, and cat were significantly decreased in the dMnMRNAi>Hand-Gal4 group (Figure 3C, D, E, F, and G).
Line 289 – 291: Sorry due to our writing error, we would like to show the following: dMnM was knocked down in Drosophila cardiomyocytes and indirect flight muscle to explore the effect of dMnM knockdown in cardiomyocytes and IFM on climbing and lifespan of Drosophila. Compared with the W1118>Hand-Gal4 group, cardiomyocyte dMnM knockdown reduced the climbing ability of the dMnMRNAi>Hand-Gal4 group (Figure 4A). However, cardiomyocyte dMnM knockdown did not affect lifespan (Figures 4B and 4C).
Line 308 – 323:
Our results described in point 3.5: regular exercise ameliorates the cardiac function defects caused by dMnM knockdown in Drosophila cardiomyocytes. Our cardiac function results showed that the cardiac cycle, arrhythmia index, fibrillation and diastolic interval were significantly reduced in the regular exercise group compared to the cardiomyocyte dMnM low expression group; whereas heart reat and shortening fraction were significantly increased. This also indicates that regular exercise can significantly improve and rescue the arrhythmia and diastolic dysfunction induced by dMnM knockdown in Drosophila cardiomyocytes. In addition, the decrease in systolic diameter without significant change in diastolic diameter suggests that the shortening fraction of Drosophila heart increases, thus improving the pumping function of Drosophila heart. Thus, regular exercise ameliorates the cardiac functional defects caused by dMnM knockdown in Drosophila cardiomyocytes.
12 Line 331 – 340 Results described in point 3.6. (Regular exercise improves cardiomyocyte dMnM knockdown leading to myocardial myogenic fiber destruction and reduction of F-Actin) seem to presented on Fig. 6 not Fig. 5, please check it.
There is no statystical analysis of these results!
Line 377 – point 3.8 - the text is missing!
Response:
Line 331 – 340: Thank you very much for your suggestion, we have fixed the writing error due to our oversight in the manuscript.we have fixed the writing error due to our oversight in the manuscript. As you said, we did not perform a statistical analysis of this result and we have recorrected the manuscript: regular exercise ameliorates the destruction of myocardial myogenic fibers caused by dMnM knockdown in cardiomyocytes. Thank you again for your careful questions.
Line 377 – point 3.8 - the text is missing! We are sorry for the omissions due to our writing mistakes. We have filled in the text of 3.8 in the manuscript. Regular exercise significantly improved the climbing ability in the dMnMRNAi>Hand-Gal4+E group than in the dMnMRNAi>Hand-Gal4 group after dMnM knockdown in cardiac myocyte (Figure 8A). However, regular exercise did not significantly affect lifespan in dMnMRNAi>Hand-Gal4+E group compared with the dMnMRNAi>Hand-Gal4 group (Figure 8B and 8C). In contrast, regular exercise significantly increased the climbing ability and lifespan of Drosophila after dMnM knockdown in cardiomyocytes and IFM compared with the dMnMRNAi>Act88F-Gal4 group (Figure 8D,8E and 8F).
13 Line 390 – 392 Sentence ‘To address this question, we used a Drosophila model to elucidate the effects of dMnM in cardiac function and regular exercise and the upregulation of dMnM expression in cardiomyocytes by regular exercise.’ must be rewritten – make it clear to the reader.
Response: Thank you very much for your suggestion, and we have changed the manuscript to read: This study aimed to illustrate the effect of low dMnM expression on cardiac function and the role of regular exercise on dMnM in cardiomyocytes using a Drosophila model.
14 Line 404 – 406 sentence dMnM knockdown allows for increased diastolic defects and arrhythmias due to prolonged cardiac diastolic intervals, and the increased diastolic interval reminds us of diastolic dysfunction must be rewritten – make it clear to the reader.
Line 413 – 414 the sentence ‘This series of results suggests that dMnM plays an essential role in the heart.’ needs a stylistic amendment.
Response: Line 404 – 406:Thank you for your suggestion, we have revised it to read:
dMnM knockdown prolonged diastolic interval and increased arrhythmias. The increased diastolic interval indicates diastolic dysfunction.
Line 413 – 414:We have completed the changes in the manuscript: These results suggest that dMnM plays an essential role in the heart.
15 Line 414 – 417 – the sentence ‘We speculate that the cause of cardiac function defects due to low dMnM knockdown in cardiomyocytes may be related to the downregulation of mRNA expression of dMnM in cardiomyocytes and the reduced anti-oxidant capacity of the heart’ is not clear, please rewrite it.
Line 421 – please rewrite this sentence to make it more clear:‘upheld that CTnT is homologous in Drosophila’
Response: Line 414 – 417: We apologize for the error in our writing and appreciate your careful reading. We have revised it to read: Cardiac function defects caused by dMnM knockdown in cardiomyocytes may be due to decreased cardiac antioxidant capacity.
Line 421: We have revised it to: upheld is cTnT homolog in Drosophila
16 Line 435 – sentence ‘Oxidative stress, defined as the excessive production of ROS relative to antioxidant levels, reactive oxygen species are oxy-chemicals with high reactivity’ needs a stylistic amendment.
Response: We have completed the changes in the manuscript: Oxidative stress in excessive production of ROS relative to antioxidant levels. Reactive oxygen species are oxy-chemicals with high reactivity.
17 Line 461 – 469 – As there is no statistical analysis of DHE staining authors cannot draw described conclusions.
Line 470 – 476 - There is no statystical analysis of discussed results! Authors cannot draw described conclusions.
Response: Response: We have performed quantitative analysis of Drosophila heart DHE staining according to your request and suggestion, and the results showed that the fluorescence intensity of heart DHE staining was brighter in the dMnMRNAi>Hand-Gal4 group compared to the W1118>Hand-Gal4 group, indicating that knockdown increased cardiac ROS production and increased cardiac oxidative stress.
Thanks again for your valuable suggestions.